# Wild Chimpanzee Welfare: A Focus on Nutrition, Foraging and Health to Inform Great Ape Welfare in the Wild and in Captivity

**DOI:** 10.3390/ani12233370

**Published:** 2022-11-30

**Authors:** Katie F. Gerstner, Jill D. Pruetz

**Affiliations:** Department of Anthropology, Texas State University, San Marcos, TX 78666, USA

**Keywords:** chimpanzees, nutritional ecology, animal welfare, macronutrients, body composition, welfare assessments

## Abstract

**Simple Summary:**

Adequate nutrition is a key factor in primate reproduction, longevity, and welfare. Thus, understanding the nutritional makeup of food choices is essential in health evaluations for wild and captive conspecifics alike. Here, we (1) highlight findings from the scientific literature on macro and micro nutritional content of foods consumed by wild chimpanzees (*Pan troglodytes*) and (2) discuss aspects of their diet, foraging activity, and health pertaining to what chimpanzees need. Additionally, although there has yet to be a standardized approach that assesses or scores individual or group welfare in wild chimpanzees, we include information from multiple study sites across the species range that is relevant to nutrition and more broadly, physical welfare in wild chimpanzees. We call for researchers to standardize welfare measures for individuals in wild populations as well. Finally, this review may be useful to captive primate managers and those who frequently look to the behavioral ecology of wild chimpanzees to inform guidelines and standards for individuals in their care.

**Abstract:**

Adequate nutrition is essential for individual well-being, survival and reproductive fitness. Yet, in wild animals, including great apes, scoring nutrition or health comes with many challenges. Here, we have two aims: first, broadly review the scientific literature regarding nutritional data on wild chimpanzee foods to get a better understanding what nutrients foods comprise of, and second, highlight important findings on wild chimpanzee nutrition and welfare pertaining to diet. We discuss variation in macro and micronutrients in food items consumed and their role in chimpanzee health across chimpanzee subspecies from multiple study sites. We found a lack of information pertaining to nutritional consumption rates of daily diets. Second, we call for a fresh, in-depth discussion on wild chimpanzee welfare issues is of foremost importance to inform conservation projects and particularly settings where humans and chimpanzees may interact, because such conversation can reveal how specific or general welfare measures can (a) inform our knowledge of an individual’s, group’s, and population’s welfare, (b) provide additional measures from the study of wild chimpanzee ecology that can guide the welfare of captive chimpanzees, and (c) can enable comparative study of welfare across wild populations. A summary of the current literature on approaches to measuring wild chimpanzee health and welfare status, to our knowledge, has yet to be done.

## 1. Introduction 

Welfare is a measurable indicator in caretaking that reflects how individuals physically and mentally cope in their environment [1]. Welfare is also a general term for state of wellbeing. In captive animal studies, behavior and physical health are two general indices that caretakers and researchers use to assess an animal’s welfare [1,2]. However, state of being is rather complex and contextual, and the facets that influence it include nutrition, sociality, space-use, illness and mental health. Describing welfare falls on a spectrum between positive and negative states. Positive welfare can indicate successful efforts and good health of animals. Whereas poor welfare can modify an animal’s behavior, biological functions, cause stress, frustration, abnormal behavior, reduce reproductive fitness, and sometimes result in death [3,4]. While approaches for assessing animal welfare in captive settings appear to be becoming a standard care practice, more applications would fit well in the realms of conservation.

The health of wild ape populations has been a focus of conservation goals, largely based on population density and demography, e.g., for heavily managed populations such as the mountain gorilla (*Gorilla beringei beringei*) where veterinary care is not uncommon [5]. Although health is an informative parameter of welfare, it can be measured from a variety of biological samples. When health is assessed with behavior observations, it does not completely portray individual wellbeing, which is less of a consideration to conservationists. For example, the welfare of chimpanzees in a community or as a species, differs from, but is informed by, the health of individuals. There has yet to be a comprehensive approach that collectively investigates, defines, or identifies the best way of measuring chimpanzee individual or group welfare in a wild setting in conservation research. This poses a challenge for today’s generation of wild chimpanzee researchers; namely, where to begin regarding welfare assessments and how these assessments can best inform conservation. As some wild primate populations become more heavily managed due to increased conservation threats, the link between using one population (captive) to inform the care and wellbeing of the other (wild) becomes more relevant (and vice versa). Overall, such welfare data is significantly valuable for informing protection policies in species conservation efforts, such as the IUCN Red List, World Wildlife Foundation, Jane Goodall Institute, and specifically can inform monitoring efforts in the early stages of care management.

As mentioned, welfare includes mental and physical welfare indicators. Physical ones for wild chimpanzees comprise information on space use, traveling, locomotion, aggression, sexual behavior, social association patterns, and foraging behavior [6]. Here, we focus on physical welfare pertaining to wild chimpanzee nutrition. Understanding chimpanzee nutritional needs and nutritional sources is fundamental to fully understanding how they live and interact in their ever-changing environments [7]. Nutrition and body composition are also strong indicators of illness and disease, and adequate nutrition is essential for chimpanzee reproductive fitness and longevity [7]. Expanding the current knowledge on what wild chimpanzees need regarding their foraging habits/behaviors, diets, and nutrition can positively inform physical welfare approaches for great apes. For this reason, we discuss below general diet, foraging and nutrition information from over 20 wild chimpanzee communities.

## 2. Diet & Nutrition

### 2.1. General Chimpanzee Diet

Wild chimpanzees are often considered omnivorous [8,9], but they are more specifically frugivores [8]. Chimpanzees will go to great lengths to find fruit, but in general, the fruit available to them is highly variable, and daily diets correlate with food item seasonality [8,9] (p. 26). In addition to seasonal fluctuations of foods, there are periods where food intake is limited because of food scarcity, and cause some degree of competition over resources within and among groups and between species sharing habitats [10] (pp. 11–12). A community’s culture is an additional important factor in chimpanzee diets, as some foods valued by one community may be completely ignored by another [11,12,13]. The breadth of a chimpanzee diet varies between populations, habitat types (forest versus savanna) as well as the level of exposure to anthropogenic factors [7]. For instance, wild chimpanzees living in savanna landscapes have diets typically low in species richness [14] in comparison to forested populations. Some populations excel at making and using tools to acquire high energy food sources like honey, nuts, and insects [9,13,14,15,16,17]. Other populations spend considerable amounts of time hunting other primates or small mammals [18,19,20].

Most wild primates have unpredictable daily diets that fluctuate in calories every day, but they typically metabolize all the caloric energy they consume through daily activities [10] (p. 23). The average male chimpanzee (*P. t. schweinfurthii*) at Gombe National Park eats roughly 1900 calories a day for normal activity levels [20]. In captive settings, chimpanzee daily energy expenditure rates, also known as total energy budget, are closer to 2400 kcal per day but needs vary with body size [21]. Due to the challenges of estimating wild chimpanzee food intake in a field setting, it is difficult to determine how much day-to-day energy consumption rates and needs fluctuate and how they compare across populations [20,21,22]. Low calorie days can have stressful and long-term effects on health, energy, and reproductive fitness [10] (p. 23). Individual factors such as age, sex, genetics, environment, and life history events can influence chimpanzees’ nutritional needs over time [23,24].

### 2.2. Foraging Activities 

Foraging and eating accounts for a large part of a chimpanzee’s life and it takes up the largest proportion of chimpanzee’s daytime activity [7]. The most common wild chimpanzee food sources eaten are fruit, leaves, insects, pith, flowers, tree cambium and honey [7,10,13,25]. There are also multiple records of chimpanzees self-medicating [26,27,28] with certain plant foods to combat gastrointestinal issues and parasites; i.e., swallowing whole plant leaves to combat tapeworm infection [26]. Impressively, chimpanzee physiology can allow for the consumption and digestion of hundreds of different plant foods [29]. Yet, selective consumption of food items by chimpanzees suggests they could be aware of their respective nutritional components [30]. In the meat-scrap hypothesis, chimpanzees at Gombe are believed to hunt for meat to obtain microminerals [18]. The concept of nutritional geometry [22,23] examines how chimpanzees balance their food intake with different nutrients and foods. It explains why chimpanzees pick the foods they do, regardless of availability, as they are selective of nutritionally balanced food items [22]. This framework requires detailed observational data on feeding behavior and applications are best suited for fully habituated populations [22,23].

Chimpanzees have evolved to withstand periods of food scarcity, which allows them to live in seasonal environments with varying food availability, to travel across large home ranges to obtain food, and compete with neighboring communities for food resources [31]. Twenty years ago, Pruetz and McGrew averaged wild chimpanzee traveling rates and found that 8–20% of their day was spent on the move [7]. Pontzer and colleagues [21] report that primates have slow metabolic rates for their body size, reproduction requirements and growth rates. Interestingly, even given the high energy expenditures suffered by wild chimpanzees, digestion rates are the same for wild and captive chimpanzees [21], and both are slower than humans. Occasionally chimpanzees are seen binging, for instance consuming excessive amounts of fruits or insects for hours [25,32]. Obesity issues have only been documented in individuals living in captivity [33]. Binging behavior in the wild might compensate for the energy expended in high-risk foraging. There are few high reward foods in the wild, and the low-risk obtainable foods with high rewards like insects or young leaves must be eaten in copious quantities to obtain significant amounts of nutrients [23,32].

### 2.3. Nutrition in the Chimpanzee Diet

The food items wild chimpanzees primarily eat fall into three distinct categories: fruits, vegetation (non-fruit plant matter) and animal matter. Fruits, dense with sugar, fiber, Vitamin C, calcium, and water are prevalent in all chimpanzee habitats and are well known to be preferred foods in the wild [6,25,29,34,35,36,37]. Vegetation including leaves, flowers, stems, bark, and pith can also provide fiber, protein, water, and other various minerals [36,38]. These items typically have longer processing times because they are fibrous and take longer time to chew or have outer layers that need removing before consuming. Animal matter (small mammals and invertebrates) is the hardest to come by. Mammalian prey may require a team effort to obtain, but it can also provide a rich reward of protein, fats, essential amino acids, and minerals that cannot be obtained from plants [20]. On the other hand, insects and other invertebrates are easier to obtain, but to get adequate amounts of nutrients, they must be consumed in large amounts, which can be time consuming [32]. Honey is harder to acquire from active bee hives [17], but it is one of the most nutrient dense foods in the wild [39].

Scientific advances in nutritional ecology research allow investigation of the organic content of many types of food items [40]. We know now that food items can be rich in multiple nutrient sources, and they can vary immensely between species, size and phenophase, such as ripe and unripe stages. Understanding the nutritional breakdown of a food item unveils its potential function(s) in chimpanzee physiology and tells us more about what a chimpanzee needs on a specific nutritional level. Macronutrient compounds include carbohydrates, protein, and lipids (Table 1). Micronutrients include a wide variety of vitamins and minerals. Below we review the nutrients and functions of common chimpanzee food items by macronutrient and micronutrients. Very few studies investigate the exact nutritional needs of wild chimpanzees, and the minimum nutritional requirements of wild chimpanzees are not known, but we reference the recent work of Uzimbabazi and colleagues [22], who averaged daily nutrition consumption rates for lactating females using nutritional geometry framework. Knowing the nutritional content of wild chimpanzee food items is a required, and Table 1. provides a sample of common foods and what they comprise of. As previously mentioned, nutritional geometry is a well-supported approach used to determine daily nutrient consumption rates [8,40], however it has limitations in field research and requires consistent observation of focal individuals, which is not always possible.

#### 2.3.1. Carbohydrates and Fiber

Wild chimpanzees eat fluctuating amounts of carbohydrates based on seasonal, habitat and resource availability [31,43]. Lactating female chimpanzees (*P.t. schweinfurthii*) at Kanyawara in Kibale National Park in Uganda, eat 76% carbohydrates (59.2–94.2%) in a 2500 calorie diet [22]. Fruit generally has a high-carbohydrate content, and fruit comprises between 60–80% of wild chimpanzee diets (Table 2) [10] (p. 23). Most carbohydrates are packed in fruit eaten in moderate to large quantities [43]. Chimpanzees prioritize it because it is easy to digest, energy rich, and palatable [43]. Researchers have observed chimpanzees travelling great distances to locate the fruit they desire, and they will seek out fruit even when it is not abundant [6,10] (p. 22).

Figs are the most common plant choice of wild chimpanzees and are present in many, if not all, chimpanzee habitats [15,16,25,29,34,35] Although they are technically not a fruit, figs get lumped into this category often in publications. Figs are inflorescences which are clumps of flowers and seeds inside bark. Chimpanzees (*P.t. schweinfurthii*) in the Kanyawara community in Kibale National Park eat ripe fruit three times more often than sympatric primates, which eat fruit more generally in other phenophases (ripe, semi-ripe and unripe) [46]. Chimpanzees here also prioritize eating ripe fruit when it is abundant [43]. At Mahale in Tanzania, chimpanzees (*P.t. schweinfurthii*) also spend most of their time eating fruit, and they focus on a small variety of species, preferring non-fibrous fruits with high caloric content and energy versus higher fiber, lower calorie, and lower protein content [37]. Several chimpanzee communities forage on cultivated fruits, which can have higher caloric value and higher water content than wild food items [36]. Studies conducted on the apes at Bossou (*P.t. verus*) in Guinea revealed these foods to have higher carbohydrate concentrations than wild fruits as well as lower fiber contents [36].

Non-fruit vegetation comprises lower calorie and lower carbohydrate food sources in wild chimpanzee diets, but they can be rich sources of fiber [23,47,48]. Plant foods including leaves, flowers, stems and shoots can be abundant sources of fiber [48]. Leaves provide considerable amounts of carbohydrates compared to other non-fruit items but much less than fruit [22]. Chimpanzees are selective with the plants they choose to eat as well as the phenophase of the plant, the part that is eaten, and the time of consumption [33,35]. At Ngogo in Kibale National Park in Uganda, chimpanzees (*P.t. schweinfurthii*) are more likely to eat sapling (young) leaves and leaves in the evening hours, when those leaf species are at their highest nutritional quality [38]. A plant food’s sugar content can differ between the morning and afternoon due to plant photosynthesis [38].

Fiber rich food sources can make up over 30% of wild chimpanzee diets [23]. Fiber is a vital nutrient for the digestion of food. Short chain fatty acids are an end product of fiber fermentation in the gut, and they are also key in maintaining a healthy gut-microbiome and energy metabolism [49]. Wild chimpanzee physiology can handle a high amount of neutral detergent fiber (digestible fiber) that is found in many of the fruits, young leaves, and flowers they consume, as well as acid detergent fibers (indigestible or complex fiber) which are found in the harder to digest plants and shoots like pith, bark, and woody stems of palm plants [25,36]. Mature leaves and exocarps of fruits usually have higher fiber contents than younger leaves and fruit flesh. Pith is the soft, fibrous flesh inside stems and palms. Chimpanzees will chew on pieces of it, a behavior called “wadging”, but then spit it out and not digest it. It is low in sugar and carbohydrates compared to fruit but can provide enough energy to get through times of fruit scarcity [36]. Short chain fatty acids are an end product of fiber fermentation, and they are vital in maintaining a healthy gut-microbiome and energy metabolism [49].

#### 2.3.2. Protein

Chimpanzees consume consistent amounts of protein in low to moderate amounts to meet amino acid requirements [23,48]. Plant protein sources such as young leaves have considerable amounts, and lesser amounts from fruits and flowers [36,37]. Like carbohydrates, plant protein content can also differ by the time of day eaten due to the photosynthesis process [38]. It appears that protein consumption is prioritized [22]. Protein content is positively correlated with food selection in wild chimpanzees as well as many other non-human primates [23]. In lactating females (*P.t. schweinfurthii*), protein broadly constitutes 4–33% of their diet (about 2500 calories) [22]. While plant protein is more likely to be available in chimpanzee habitats, animal matter is more protein dense.

Even though insectivory varies between field sites, almost all studied chimpanzee communities consume insects opportunistically at the very least [13]. At Gombe, 56% of females’ fecal samples contained at least one type of insect, versus only 27% in males [41]. Termites, such as *Macrotermes* sp. and *Dorylus* sp. are consistently high in protein [41,50] and contain comparable amounts of protein to the reported contents of vertebrate prey species. Chimpanzees are also selective when it comes to insects and show a strong preference for certain genera while ignoring available ones that may be favored by other populations [51]. In Uganda, the absence of termite consumption may be due to taste, diet, or a lack of knowledge of using extractive tools [52].

Animal meat is another high protein source, but consumption varies among individuals and is influenced by rank, skill and other factors [53]. Meat is sometimes shared (or theft is tolerated) among multiple individuals because many hunts usually involve multiple individuals [52]. Not all individuals receive the same amounts of protein in this way. Pieces vary in nutrition composition [19,20,54], and since hunting is an opportunistic activity, chimpanzees do not get the same amount of protein every day or any at all from meat.

#### 2.3.3. Lipids

Lipid sources are limited in wild primate environments including those of chimpanzees, nevertheless chimpanzees like fat dense foods, which are filling, calorie dense, and can help provide energy in times of deprivation or when food is not abundant. Lactating females eat on average 7% lipids (0.5–19.0%) in a 2500 cal diet [22]. Vertebrate and insect foods are primary fat sources and contain the highest amounts among chimpanzee food items [8] depending on the part consumed and species [20]. Mammalian brains and livers contain high lipid concentrations and were most often the first targeted area for consumption by chimpanzees at Gombe (91% of 29 captures) [20]. Trivial amounts of fat are acquired through the consumption of insects, and amounts are highly variable between hard and soft bodied species and species age [8,50].

Plant foods can also contain lipids [36,37,38]. Nuts contain considerable amounts of fat, but similarly to animal matter, consumption rates and species availability are variable between populations and seasonality [35,46]. Seeds from fruits and flowers also contain lesser amounts of lipids but can provide consistent portions of daily fat targets when consumed in bulk [23]. Fongoli chimpanzees (*P. t. verus*) reingest digested baobab (*Adansonia digitata*) seeds [30]. Baobab seeds are fat dense and the nutrients, like fat are easier to acquire after being digested once [30]. At Bossou, nutritional analysis revealed high fat contents in cacao fruit, palm oil fruit and palm oil nuts [36]. Cacao and palm oil are cultivated foods at this site and have at least a third more fat content than any of the native fruits (from this site) known to contain significant amounts of fat [36].

#### 2.3.4. Vitamins & Minerals

There has been a growing amount of research on non-fruit food items rich in microminerals. Foods such as insects [32,41], crustaceans [55], pith [56], soil [57], and decaying wood/bark [58,59] contain essential microminerals, including calcium, phosphorous, magnesium, potassium, sodium, chlorine, sulfur, iron, zinc, copper, manganese, iodine, cobalt, selenium, and chromium because research on captive nonhuman primates found that they are imperative for physiological processes [48]. In a wild community, at Budongo Forest in Uganda, a micromineral consumption summary reported chimpanzee (*P.t. schweinfurthii*) diets are high in calcium, moderately high in potassium and magnesium, and lack aluminum, copper, and iron [57]. Chimpanzee vitamin and mineral requirements vary between wild populations most likely due to habitat type and resource availability. Table 3 gives a broad overview of food items that are rich in microminerals from field sites in East and West Africa.

Termite prey species are dense with microminerals, fatty acids (oleic acid and linoleic acid) [50], and vitamin B12 [32,50] but rates vary per species [32]. As previously mentioned, chimpanzees show regional differences in the animal species they eat. Foraging for animal prey items could emphasize the need for certain microminerals [20], or it could show a preference due to palatability [57]. It is difficult to estimate how much is minimally required because chimpanzee populations may consume animal prey at different rates, seasons, or not have them available at all. From published data on chimpanzee termite consumption, only eight of 85 available genera of termites have been observed to be eaten [60].

Some food items with considerable amounts of sodium are animal prey and ripe fruits [20,32,37], followed by pith and decaying wood [56,58,59]. Sodium is essential in maintaining and regulating water intake in the body and iron is essential for transferring oxygen in the circulatory system. Mammal and insect prey are generally high in iron and sodium [20,32,37]. Chimpanzees can hunt for chances to obtain iron while dependent infants acquire it through breast milk [54]. Insects can also provide significant amounts of iron when consumed in large quantities, but contents vary per species [32]. Geophagy, or soil eating, by chimpanzees (*P. t. schweinfurthii*) of the Sonso community at Budongo Forest provides micronutrients like iron [57]. Sonso chimpanzees also drink clay-infused water which contains a significantly high amount of iron and researchers propose this behavior can help remove organic chemical compounds from their diet [57].

## 3. Health Assessments

Ultimately, data on diet and nutrition informs us regarding the health of an animal but linking this information to aspects of chimpanzee behavioral ecology that figure into fitness in an evolutionary context is challenging. This link is instrumental to an evolutionary understanding of wild chimpanzees and therefore captive chimpanzee management. Bringing health into focus for wild apes, as is done for captive chimpanzees, can help fill this gap.

There is currently not a standardized way to measure chimpanzee nutrition or welfare across study sites of wild populations. As mentioned, the literature lacks a clear definition of what “good” health is in the wild. While there is a guidebook on the best practices for monitoring health and diseases in great apes [61], it is not clear if all chimpanzee field sites collect long term data on chimpanzee health or enough data needed to estimate nutrition. Documenting health is an increasing trend and a wise one, considering the increased amount of research supporting disease transmission between humans and chimpanzees [61,62,63,64]. Noninvasive methods like the collection of biological samples such as fecal matter, urine or hair, biomarker identification, and DNA extraction techniques can determine levels of hydration [65], stress [65,66], parasite infection [67,68] or virus densities [69] Even though these methods are extremely insightful, the downside is that they can be costly and time consuming, with results taking weeks to months. Furthermore, many field sites do not have the means to conduct laboratory experiments on site and not every university is outfitted with the equipment to conduct such analyses. It is common for researchers to transport samples back to their affiliated institutions for this type of testing, which again takes time.

Select long-term chimpanzee monitoring efforts use site specific protocols for assessing health through behavioral observations. Lonsdorf and colleagues [68] shared information on Gombe National Park’s health-monitoring protocol, which began in 2004 with the aim of understanding the baseline health levels of wild chimpanzee health. Researchers here document clinical health signs that correlate with infections such as poor body condition, and abnormal behavior including lameness or lethargic behavior. Presence of illness symptoms are recorded, including coughing, running nose, sneezing, watery fecal condition, and occurrence of wounds or injury. Consistent data collection and recording observed clinical health signs and collection of biological samples for further testing is a more reliable way to confirm the presence of illnesses and diseases.

Negrey and colleagues [69] shared that Kanyawara and Ngogo population researchers now record daily the presence or absence of clinical health signs using a health scoring system. This health scoring system has categories for the presence of coughing, sneezing, diarrhea or abnormal feces, presence of unhealed wounds, skin abnormalities or swelling, and mobility deficits such as limping. Researchers consider chimpanzees to be ill if they display one or more of these signs.

At Fongoli, chimpanzee health and wounding has been recorded systematically since habituation in 2005 based on captive protocols used by the Keeling Center in Bastrop, TX, USA (Fongoli Savanna Chimpanzee Project, unpublished data). In addition to females’ reproductive state (i.e., estrous score), any wounds or illnesses are recorded daily/when individuals are encountered. Symptoms of illness, such as diarrhea or loose stools, sneezes, coughs and descriptions of the healing progress of wounds, as well as tooth wear (or breakage), skin depigmentation and pelage color change in mature individuals are included in these records. Medicinal plant use as well as the appearance of parasites in stools are recorded opportunistically at Fongoli, with more systematic research, including study of parasite load, occurring periodically.

While uniform health scoring systems have potential, there is also a need for standardized protocol for assessing nutrition. Though records of nutritional deficiency related illnesses are rare in the wild, nutrition is a key aspect of animal welfare. Scaling body composition is one way to examine weight gain or loss. Assessments can be done quickly and noninvasively through clear observations of individual direct observation or through camera trap footage. At Fongoli, qualitative descriptions of body composition have been used because such assessments were more informative in the rare cases where an individual was ill, geriatric or wounded. Although this method may be suited for small populations, a scaling system would be more practical to use for larger chimpanzee populations and for systematic comparisons to be made across sites. Reamer and colleagues [70] implemented body composition scoring with a 10-point scale for assessing emaciation and obesity in retired former research chimpanzees in captivity at the Keeling Center. The body composition scoring method allows care teams to rate individuals noninvasively and has proven valid when compared to individual weight and assessment data collected during routine exams while sedated. The study includes a photo table of body variations per score that were created by veterinarians based on chimpanzee physiology and sex differences. A body composition scoring system could be implemented in the increasingly prevalent number of studies conducted on unhabituated wild chimpanzees using camera trap data.

Although a standardized methodology for scoring wild chimpanzee welfare does not exist, a suitable assessment called the Five Domains Model was recently applied to free-ranging horses [71]. Originally a protocol designed to score the welfare for research and laboratory animals in the mid 1990s, the Five-Domains Model is a device that facilitates systematic and structured welfare assessments on individual animals by linking their physical and mental states and is based on the functional basis of positive and negative experiences [71,72]. With horses, Harvey and colleagues [71] provide step by step protocols for assessing nutrition, environmental, health and behavior of individuals in wild settings, in addition to protocols for investigating their connections to individual affective experiences. The results were used to inform conservation efforts for large mammals in Australia. This recent research demonstrates how the model is flexible and malleable to be species-specific and population specific with the right multidisciplinary input and extensive knowledge [73].

## 4. Conclusions

The information provided here can help captive managers and field researchers better understand the components of wild chimpanzee nutrition and health. Given the increasing rate of habitat fragmentation of primate habitats, indirect nutritional information can continue to inform what chimpanzees need in their diets. We hope future comparisons on wild chimpanzee nutrition and welfare will lead to new advances in the conservation and identification methods for wild food items, in addition to a standardized approach for assessing wild chimpanzee welfare and continue to guide the care of captive chimpanzees.

## Figures and Tables

**Table 1 animals-12-03370-t001:** Published macronutrient composition of representative common wild chimpanzee food items from sites in East and West Africa (Gombe and Bossou, respectively). Amounts reported are averages. Exact amounts will vary between species and location. Nutritional data was sourced from analyses conducted on chimpanzee food items in wild habitats (g/100 g). Nutritional information on honey from chimpanzee habitats is lacking, and provided data is from a study on Hadza foragers from Tanzania.

	Macronutrient				
Food Item	Carbs	Fat	Crude Protein	Total Calories	Fiber
Figs (*Ficus* sp.) [36]	17.74 (g)	0.5 (g)	10.3 (g)	246 (kcal)	59.41 (g)
Non-fig fruit (*Psuedospondias microcarpa*) [36]	37.46 (g)	4.64 (g)	9.78 (g)	293 (kcal)	38.81 (g)
Termite (*Macrotermes* sp.) [41]	1 (g)	31.2 (g)	19.7 (g)	369 (kcal)	0
Honey * *(species combined)* [39]	91.6 (g)	4.3 (g)	2.9 (g)	420.8 (kcal)	0
Young leaves *(species combined **)* [42]	23.3 (g)	0	25 (g)	277 (kcal)	2.7 (g)
Pith (*Aframomum latifolium*) [36]	37.46 (g)	5.15 (g)	6.7 (g)	205 (kcal)	0.66 (g)

* Honey values are a combined total average of honey extracted from multiple hives in different locations. ** Young leaf values are a combined total average of all leaves foraged in a year by chimpanzees at Bossou, Guinea.

**Table 2 animals-12-03370-t002:** Published estimates of diet composition in different chimpanzee communities. Diet is broken down into fruit, vegetation and animal matter consumed across different chimpanzee communities. Annual diet compositions (%) of each food category. Animal matter consists of mammal and insects. The percentage of animal matter in Ngogo diets was not found, but they do hunt more often and have high success rates compared to other large communities, which be may due to larger party sizes [44].

Chimpanzee Community	Fruit	Leaves	Animal Matter	Subspecies
Mahale [37]	64.5%	5.8%	<5%	*P. t. schweinfurthii*
Gombe [9]	63%	19%	<5%	*P. t. schweinfurthii*
Ipassa [35]	68%	28% (leaves and stems)	4% (2.5–6%)	*P. t. troglodytes*
Budongo [15]	64.5%	28.5% (leaves and flowers)	<3%	*P. t. schweinfurthii*
Ngogo [29]	70.7%	19.6%	--	*P. t. schweinfurthii*
Fongoli [16]	62.5%	16%	0.5%	*P. t. verus*
Tai [45]	84.5%	11%	3.3%	*P. t. verus*

**Table 3 animals-12-03370-t003:** Chimpanzee foods rich in microminerals. computed into average amounts or listed as percentages of daily intake, depending on the study’s methods. Food items were sampled from studies conducted at Bossou, Guinea and Gombe, Tanzania.

	Micromineral				
Food Item	Sodium	Potassium	Calcium	Iron	Magnesium
Decaying Wood (*Cleistopholis patens*) [57]	1871 (mg/kg^−1^)	10,816 (mg/kg^−1^)	5682 (mg/kg^−1^)	148 (mg/kg^−1^)	2136 (mg/kg^−1^)
Decaying Wood (*Neoboutania macrocalyx*) [58]	3203 (mg/kg)	9478 (mg/kg)	4221 (mg/kg)	141 (mg/kg)	2240 (mg/kg)
Termite Soil (6 species) [56]	5–47.1 (mg/kg)	1080–1980 (mg/kg)	1030–3270 (mg/kg)	49,100–32,100 (mg/kg)	--
Decaying Pith (*Raphia farinifera*) [55]	5738 (mg/kg^−1^)	6650 (mg/kg^−1^)	1563 (mg/kg^−1^)	128 (mg/kg^−1^)	2430 (mg/kg^−1^)
Termite soldiers (*Macrotermes subhyalinus*) [50]	13.8 (mg)	79.4 (mg)	6.9 (mg)	7.6 (mg)	6.9 (mg)

## Data Availability

No new data were created or analyzed in this study. Data sharing is not applicable to this article.

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
