# Peer review of "Wild Chimpanzee Welfare: A Focus on Nutrition, Foraging and Health to Inform Great Ape Welfare in the Wild and in Captivity"

_animals, 2022, doi:10.3390/ani12233370_

Round 1

Reviewer 1 Report

This article is very informative as it contains information about the diet of wild chimpanzees in one review article.  However, the introduction, discussing “positive welfare” sets-up the reader to expect something different (I think of welfare as embracing both physical and mental health - but it seems the focus of the article is physical health).  I would like more suggestions about how this information about wild diets can be utilized by those in captive management.  Wild diets can be so variable, across seasons, across ecological settings, across sex, and so are the authors suggesting that captive management teams mimic this?  I wondered about how this information can best be used for implementation in captive settings.

Please provide a definition of “welfare” and specify how it differs from health, in terms of nutrition.

Sometimes, I felt that the authors were trying to say that we should be assessing the [physical] welfare of wild groups.  The final section was useful in specifying some ways that this could be done, but I was not clear about why the authors want to do it.  Are they thinking of interventions for wild chimpanzee populations? Or is that the only way that one can then appropriately use the dietary information from wild chimpanzees (i.e. knowing that that diet supports good health)? However, the conclusions seem to focus on captive groups and the importance of nutrition for them.  I am not quite sure what this problem is, but there is a lack of clarity or balance between the wild information and the applicability for captive groups.

Please provide more detail of the Five-Domains Model – is this actually used in the current paper?  I am not sure why it is presented in the current somewhat cursory fashion.

Please be sure to include information about chimpanzee diet from all long-term field sites (it seems that Mahale Mountain is less cited, as is Tai Forest).

Since the authors remind us that wild populations can have parasites, undiagnosed diseases, and be “on the edge” of malnourishment, I wonder how we are to evaluate applying wild diets to captive chimpanzees.  It was with this in mind, that I thought the providing a table with variability in diets might be informative for the reader.

The main text is a bit dry – I wondered if the authors might consider cross-site variations within each section – perhaps with mortality data – to give a picture of chimpanzee health that embraces the diversity across ecological settings.  It might also be interesting to include some indication of the current ‘best practice’ for chimpanzee diets in captivity as a point of comparison. (e.g., surely captive chimpanzees do not need 2500 calories per day).

Some stylistic suggestions:

Some statements throughout the ms have no empirical anchor, e.g., chimpanzees are selective in their carbohydrate food choices, or “protein consumption is prioritized”, or foraging is a large part of a chimpanzees life.  It would help the reader to have these types of statements accompanied by percentage of day time or percentage of diet, etc.

I think, in general, there is lack of explicit acknowledgement that certain foods can provide both carbohydrates and fiber, for example.  Thus, I wondered if it might be useful at some point later in the ms, to say approximately what is eaten in a day/season (how many fruits, how many leaves, how many insects) and how much of waking time is spent eating by wild chimpanzees.  It seems that one of the major problems for captive management is to approximate wild foraging times – especially if the management strategy is to maximize dietary needs in single packages (e.g., primate biscuits).

It was not clear to me how much of the information suggests dietary “requirements” and how much is either recommended or simply acknowledged as present in wild diets – this is especially relevant to the micronutrient section.

I liked the information presented in Table 1.  I recommend that additional tables be included (e.g., perhaps as suggested above to highlight variation across sites).

The added information on the difference between cultivated fruits and wild fruits was interesting.  This adjustment then should clearly be taken into consideration for any fruit in the diet of captive chimpanzees (since presumably it would be all cultivated).

In conclusion, I liked the paper.  Some tightening of the focus will help the readers receive the important messages therein.

Author Response

  • Clarified welfare terms in the introduction
  • Emphasized food item variation across study groups
  • Clarified the connection between this study's information and it's relevance to captive care
  • Provided more detail for the 5 domains model and relocated this section to the discussion
  • added tables on food item variation 
  • rewording and rephrasing throughout revision

Reviewer 2 Report

Dear authors,

I believe that a review should provide more detailed and in-depth information on the field to be addressed. Next, I put some suggestions with the aim of improving the work.

According to the journal’s regulations, reviews must have at least two tables and/or figures.

Who is the corresponding author? The institutions look the same, yet they are like two different ones.

The work does not have the number of the lines, this makes it more complicated to review.

Keywords should not be in bold.

Table 1. The composition of micromines is calculated or analyzed? This should be specified in the Table. Also. This nutritional information is expressed in dry matter or in fresh matter. Some of them, such as soldier termites, are expressed in mg, but I understand that it will be in mg per kilo or per gram. All this must be well explained. Please remember that both figures and tables must be self-explanatory.

It appears in the Abstract that it is a review of macronutrients but the table is of micronutrients, specifically microminerals.

There is no clear explanation of energy, which is normally the most important and limiting nutrient. The second should be the protein.

They talk about carbohydrates and fiber together. There is no explanation of van soest fibers, for example. They should be studied separately.

In the review of the protein part there is no clear and concise information on the essential amino acids, if there is any that may or may not be limiting.

Regarding this line, protein broadly constitutes on average 15% of an average 2,500 calorie diet (4-33%) % of what, of calories? Would it be expressed per kcal?

Regarding protein, “is positively correlated with food selection in wild chimpanzees as well as many other non-human primates [50]”. How does this relate to not having great needs?

I advise you to carry out a methodological study of the nutritional needs and/or the nutritional assessment of the main foods of these animals. The basics of nutrition are based on this and it is what you would expect to read in a review. After talking about the nutritional requirements, they could talk about the assessment of plants and animals, and not only mark the micronutrients, but do it with all the chemical components.

All the best,

Author Response

  • revised version added 2 additional tables
  • Revised formatting errors (bolding keywords, editing institutions, adding page lines) 
  • elaborated specifics on nutrients, protein, fiber, and caloric ranges

Reviewer 3 Report

The paper “Wild Chimpanzee Welfare: A Focus on Nutrition, Foraging and Health Assessments.” attempts to review the nutritional requirements of wild chimpanzees to better assess chimpanzee welfare in the wild and captivity. Review papers are notoriously hard to write due to the need for exhaustive research. I congratulate the authors’ efforts and extensive research focusing on all major nutritional requirements of wild chimpanzees, although they perhaps overly focus their arguments on a few references (e.g., Stanford (2018), and Uzimbabazi et al. (2021)). The paper is relevant not only because it can help better the welfare of captive chimpanzees, but also by providing important information that will help field researchers to assess nutritional and overall health of wild chimpanzees. This last point is of particular relevance to the field of primatology given the increasing number of chimpanzee communities inhabiting highly fragmented habitats, in close proximity to human populations.  

One of my main concerns with this paper is the lack of clear goal(s).

From the title it appears that the main focus of this study is wild chimpanzee welfare. In the simple summary, the main objectives are to use our knowledge of wild chimpanzee nutrition to help care for (captive?) chimpanzees, as well as to use protocols from captivity to help assess wild chimpanzee welfare. In the abstract, the main goal is to study wild chimpanzee nutrition to assess good welfare in the wild as well as use wild chimpanzee behavior to assess captive chimpanzee welfare and create protocols to assess wild chimpanzee welfare. Finally, in the Introduction the main goal is to use information from the wild to create guidelines for captivity.

Is the main goal to learn how to assess wild chimpanzee welfare based on nutritional studies in the wild, and/or captive studies?

Or is the main goal to assess captive welfare by studying wild chimpanzee nutrition?

Are all of these the goals of the paper? Please make the goals clear in all relevant sections and make sure that throughout the paper there is adequate and thorough discussion of both wild and captive welfare/health/nutrition.

Secondly, I would like to see a more in-depth analysis of what welfare actually means.

The authors include on page 2 a short definition “Welfare is a measurable indicator that reflects how individuals physically and mentally cope in their environment [6]. Behaviour and physical health are two general indices that caretakers and researchers use to assess an animal’s welfare, or state of being [5,7].”  However, given that this paper has a focus on welfare, I would like to see a bit more on it, especially the reasoning behind focussing on diet and nutrition, and what welfare in the wild actually means.  

Moreover, the conclusion appears to be somewhat disconnected from the rest of the paper. The paper focusses mainly on wild chimpanzee nutrition and some health assessments done in specific wild chimpanzee sites and a few captive ones, however the conclusion focusses mostly on captive chimpanzees.

Finally, I believe the English needs to be polished throughout with a specific emphasis on the section on nutrition.

I have added detailed comments, questions, and suggestions directly to the document (attached).

Author Response

  • We polished the English and modified the language used throughout the manuscript as suggested in the attached pdf of revisions. Thank you, reviewer for kindly putting your time into these edits.
  • We solidified paper goals, the main goal being to review wild chimpanzee food item nutritional content and the second goal is to review relevant literature pertaining to wild chimpanzee health and welfare. 
  • We defined welfare and it's factors at the beginning of the introduction. 
  • In the discussion, we added information pertaining to the 5 domains model and removed the captive mobility assessment study, as suggested. 

Round 2

Reviewer 2 Report

The authors have made the suggested changes.

Author Response

We thank the reviewer for their time and feedback on the manuscript. 

Reviewer 3 Report

The authors provided some answers to my comments, but I would have like to have seen more detailed answers and justifications for certain comments, such as the over citation of two references. Additionally, several of my comments and corrections appear to have been forgotten or ignored.

It appears that the review was done in a rush, since there are several sentences that either don’t make much sentence, are not finished, or even repeated, despite my previous comments about the English needing to be polished. There are elements missing in tables (such as scientific names, or the ones shown are incorrect), the reference numbering also has several errors both in the text and reference section.

Please see my further comments bellow.

Simple Summary:

First sentence “ … and guide regarding relevant to assessing “ This doesn’t make sense, please rephrase/edit

L 13 this sentence should be the one starting with “finally” since this appears to be the final point that the authors want to make.

Abstract

L19-20 “ one component of physical welfare: nutrition and health measures, indicators, and necessities.” It starts by saying it is one component but then goes on to give a list. Please rephrase.

L20-22 Confusing sentence, please rephrase

L22 “range in macronutrients of food items” should read “ range of macronutrients in food items”

L23” that could be” should read “that can be”

Introduction

L43 “where decades of ongoing research inform promoting the most” this sentence doesn’t make sense, please rephrase.

L44-46 is there a reference?

L58-60 “the link between captive and wild ape welfare, and using the populations to inform the care and wellbeing of the other becomes more relevant. “confusing, please rephrase

L58 Please explain what “heavily managed” means in the context of wild populations

L64 “includes mental and physical welfare indicators and physical ones” rephrase, maybe divide into two sentences

L72 “foraging” should be foraging habits/behaviour

L71-73 needs to be better explained

L73-78 these sentences are repetitive, please simplify and combine

L87 “population’s culture “ should be a community’s/group’s

L96-97 rephrase

L102 delete full stop before reference

L107 this example of female chimpanzees hunting is site specific, no? make it clear

L122 instead of “populations” maybe “communities” would be best, same for the next line

L 123 the point of this sentence is to show that there are now studies done in different communities and subspecies, however the authors use only one reference that turns out to be for eastern chimpanzees. Please add relevant references from different sites and subspecies

L126.”live in seasonal” something is missing

L130 if there are examples of obesity in the wild? If so, please reference; if not rephrase.

L137 delete  “of”

L137 – paragraph needed

L160 “novel food items” which ones are the novel food items, does this mean newly introduced into the habitat? Why were these the items chosen for this table?

Table 1 – I don’t quite understand how these foods were selected and why. A better explanation is needed. Additional, why doesn’t tamarind have a scientific name?  The scientific name for Palm oil is wrong, “Citrus sinensis” is orange. Why don’t the young leaves have a scientific name? The data shown on this table is adapted from other papers, this needs to be explained in the table’s caption with references. But given that this is a review it would have been better to add data from more studies.

L161. This is a comment I had before. If there are few studies these should all be referenced. If then the authors chose to only work with the data of Uzimbabazi this should be explained and justified.

 L173 do you mean “resource availability”?

Table 2  - Should read  “Variation of fruit, vegetation and animal matter” consumed  “across different chimpanzee” communities. Are there no other papers with meat consumption (e.g. Tai)? And specially of chimpanzees of different subspecies?

L185- either use “fig” or “ficus” and stick to it throughout

L186- is this “although” meaning this community doesn’t feed on figs? Needs rephrasing

L189- should be “chimpanzee communities” and “ forage on cultivated fruits” or “forage on crops*  

L203 this sentence was left unfinished…

L223- please rephrase

L228 I think this reference is not the right one, since it is about captive chimps…

L247 “provide their bodies” with what?

L253 “tiny” ?

L 262 give scientific names

L263 oil palm is native to Africa, it can be considered a crop on specific chimpanzee sites but not in others

L268-271 and L274-277 are the same

Table 3 – Again it needs to say in the caption that this table is adapted from other work

L289 “Between all the studied chimpanzee communities” like I previously explained most studied chimpanzee communities have been studied via indirect methods which means that in many cases we cannot confirm the consumption or not of certain foods. You cloud say instead “ From published data on chimpanzee termite consumption only 8 out of 85 termite species have been confirmed to be eaten” or something similar. I assume the number of this ref is also incorrect

L293 delete full stop

L296 “intake” or “levels” ?

L300 it is repeating information from L298

L318 missing a “]”

L362 I think you mean “emaciation” not “emancipation”

L364 should read “and has proven valid when” or “and its results have proven valid when”

L365 “For the best reliability of scores, individual bias can be a factor” needs to be rephrased

L369 “tap” should be “trap”

L371-382 this paragraph appears to be an afterthought, needs to be better connected to the rest of the text  

L370 give some references

L384 this is not the best way to start a conclusion…

L384-386 “not only because” suggests that there is more than one reason… something is missing in this sentence

L386 what “last point” ?  I’ve just noticed that this part of the conclusion is copied word by word from my comments… I’m not one of the authors.

"This review is relevant not only because it can help provide valuable information that will help field researchers to assess nutritional and overall health of wild chimpanzees. This last point is of particular relevance to the field of primatology given the increasing number of chimpanzee communities inhabiting highly fragmented habitats, near human populations." (L384-387)

vs 

"the paper is relevant not only because it can help better the welfare of captive chimpanzees, but also by providing important information that will help field researchers to assess nutritional and overall health of wild chimpanzees. This last point is of particular relevance to the field of primatology given the increasing number of chimpanzee communities inhabiting highly fragmented habitats, in close proximity to human populations.  " (my initial comments) 

References

The numbers of the references need reviewing. There are several references with #1 on the first and second page
